# Co-Aggregation of S100A9 with DOPA and Cyclen-Based Compounds Manifested in Amyloid Fibril Thickening without Altering Rates of Self-Assembly

**DOI:** 10.3390/ijms22168556

**Published:** 2021-08-09

**Authors:** Lili Arabuli, Igor A. Iashchishyn, Nina V. Romanova, Greta Musteikyte, Vytautas Smirnovas, Himanshu Chaudhary, Željko M. Svedružić, Ludmilla A. Morozova-Roche

**Affiliations:** 1Department of Medical Biochemistry and Biophysics, Umeå University, SE-90781 Umeå, Sweden; l.arabuli@ug.edu.ge (L.A.); igor.iashchishyn@umu.se (I.A.I.); nina.romanova@umu.se (N.V.R.); 2Department of Natural Sciences, School of Science and Technology, University of Georgia, 0171 Tbilisi, Georgia; 3Institute of Biotechnology, Life Sciences Center, Vilnius University, LT-10257 Vilnius, Lithuania; gm629@cam.ac.uk (G.M.); vytautas.smirnovas@bti.vu.lt (V.S.); 4Department of Biotechnology, University of Rijeka, HR-51000 Rijeka, Croatia

**Keywords:** amyloid, binding, cyclen, DOPA, morphology, S100A9

## Abstract

The amyloid cascade is central for the neurodegeneration disease pathology, including Alzheimer’s and Parkinson’s, and remains the focus of much current research. S100A9 protein drives the amyloid-neuroinflammatory cascade in these diseases. DOPA and cyclen-based compounds were used as amyloid modifiers and inhibitors previously, and DOPA is also used as a precursor of dopamine in Parkinson’s treatment. Here, by using fluorescence titration experiments we showed that five selected ligands: DOPA-D-H-DOPA, DOPA-H-H-DOPA, DOPA-D-H, DOPA-cyclen, and H-E-cyclen, bind to S100A9 with apparent K_d_ in the sub-micromolar range. Ligand docking and molecular dynamic simulation showed that all compounds bind to S100A9 in more than one binding site and with different ligand mobility and H-bonds involved in each site, which all together is consistent with the apparent binding determined in fluorescence experiments. By using amyloid kinetic analysis, monitored by thioflavin-T fluorescence, and AFM imaging, we found that S100A9 co-aggregation with these compounds does not hinder amyloid formation but leads to morphological changes in the amyloid fibrils, manifested in fibril thickening. Thicker fibrils were not observed upon fibrillation of S100A9 alone and may influence the amyloid tissue propagation and modulate S100A9 amyloid assembly as part of the amyloid-neuroinflammatory cascade in neurodegenerative diseases.

## 1. Introduction

Amyloid formation is a widespread phenomenon based on the generic property of the polypeptide chain to self-assemble into a cross-β-sheet containing oligomers and fibrils [1,2]. Their growth and accumulation are manifested in numerous amyloid-related diseases [3,4,5], including neurodegenerative diseases, such as Alzheimer’s and Parkinson’s. Despite the key clinical significance of amyloid formation, the mechanisms of its inhibition, reversal, and modification remain elusive. The cross-β-sheet structure at the core of amyloid fibrils is stabilized by the numerous hydrogen bonds of the polypeptide backbone. In addition, π-π stacking of aromatic residues can also contribute to amyloid self-assembly and stability [6]. Small phenolic compounds alone or conjugated with various groups were found to be effective in targeting the monomeric polypeptides as well as their aggregates. They have shown potential activity in animal models of Parkinson’s disease, and some have already entered clinical trials [7,8,9,10].

Here, we consider the effect of cyclic compounds and their conjugates on the amyloid formation of pro-inflammatory S100A9 protein, which was found to be a common denominator in Alzheimer’s and Parkinson’s diseases as well as in traumatic brain injury, which is considered to be a pre-cursor state for neurodegenerative ailments [11,12,13]. Indeed, amyloid formation is commonly associated with neuroinflammation, and pro-inflammatory S100A9 protein acts both as an alarmin, inducing the production of pro-inflammatory cytokines, and as a highly amyloidogenic protein, which self-assembles into amyloids under physiological conditions [14,15]. By combining in vitro, ex vivo, and in vivo studies, we have demonstrated that S100A9 may drive the amyloid-neuroinflammatory cascade in neurodegeneration both in humans and in a mice model [11,12,13,16,17]. Moreover, S100A9 co-aggregates and forms joint complexes with major amyloid polypeptides, such as amyloid β (Aβ) peptide in Alzheimer’s disease [11,18] and with α-synuclein in Parkinson’s [12]. This would imply that upon altering the aggregation pathways and final structures of one component of the amyloid cascade, other components and the whole cascade could also be modified.

Furthermore, growing evidence demonstrates that apart from disease-associated amyloids, there are also functional amyloids, which play useful roles in numerous biological processes in many organisms, ranging from mammals and insects to fungi and bacteria [19,20]. For example, peptide and protein hormones can be stored in an inert amyloid state in secretory granules of the endocrine system [21], transcription and translation can be regulated by prion proteins in yeast [22], spidroins enhance spider web tensile strength, and bacterial functional amyloids, such as curli and FapC, participate in the formation of bacterial biofilms [19]. In recent years, the possibility of cross-seeding of disease-associated amyloids by functional bacterial amyloids has received increasing attention as this mechanism can be linked to the gut–brain axis of neurodegeneration disease development upon changes of the gut microbiome [23,24]. It has been shown that the aggregation and toxicity of Aβ, the pathogenic peptide associated with Alzheimer’s disease, can be seeded by FapC amyloid fragments of *Pseudomonas aeruginosa*, which colonizes the gut microbiome through infections [24]. Thus, the link between the gut microbiome, their metabolites, and neurological disorders, such as Alzheimer’s and Parkinson’s [23,24], suggests an additional pathway that attunes amyloid formation of disease-related proteins via modulation of amyloid self-assembly of other amyloid counterparts and their amyloid seeds.

To target S100A9 and its amyloids, we developed conjugated molecules by combining cyclic compounds, which may potentially target aromatic residues and hydrophobic regions within the amyloid structures, with charged di-peptides, in order to make the hydrophobic moiety more soluble. One group is based on L-DOPA—L-3,4-dihydroxyphenylalanine (DOPA), which is an amino acid that is made and used as part of the normal biology in humans as well as in animals and plants. DOPA is produced from the amino acid L-tyrosine by the enzyme tyrosine hydroxylase and can act as an L-tyrosine mimetic to be incorporated into proteins in place of L-tyrosine, generating protease-resistant and aggregation-prone proteins [25]. Sufferers from Parkinson’s disease are exposed to DOPA over many years as a therapeutic agent for their medical condition as it serves as a pre-cursor for the catecholamine neurotransmitter dopamine. Dopamine is produced in the brain and plays several important biological roles, acting as both a neurotransmitter and hormone [25,26,27]. Dopamine production and homeostasis in neurons is regulated by α-synuclein through the interaction with tyrosine hydroxylase. It has been reported that dopamine can form adducts with α-synuclein in vitro, which stabilize the α-synuclein protofibrils and inhibit its fibril formation, while certain dopamine derivatives alleviated α-synuclein-engendered defects in a Parkinson’s animal model [28,29,30,31,32]. Naphthoquinone–DOPA hybrids inhibit α-synuclein and tau aggregation, disrupt preformed fibrils [33,34], and attenuate aggregate-induced toxicity [33]. DOPA and DOPA-conjugated naphtalenediimides also modulate Aβ toxicity [35]. Moreover, recent studies have reported the self-assembly of DOPA-containing building blocks into nanometric fibers [36,37], characterized by a unique functionality [38]. In addition, a short synthetic pentapeptide, containing two DOPA moieties, D-DOPA-N-K-DOPA, retained the ability to spontaneously self-assemble into amyloid-like fibrillar assemblies in water, and those assemblies displayed structural properties characteristic of amyloids [39]. Here, we linked DOPA to negatively charged aspartic acid residue (D) and positively charged histidine residue (H) to produce the following soluble compounds: DOPA-D-H, DOPA-D-H-DOPA, and DOPA-H-H-DOPA (Figure 1).

In addition, DOPA was linked to saturated macrocyclic polyamine cyclen—1,4,7,10-tetraazacyclododecane (cyclen), which belongs to a class of nitrogen-containing heterocyclic compounds, and DOPA-cyclen and H-E-cyclen were used here to compare their effect on S100A9 fibrillation vs. DOPA-based conjugates. Macrocyclic polyamines, including cyclen, have been very broadly used in medicinal chemistry as therapeutic or diagnostic agents [40,41] as well as to target amyloid formation via chelating metal ions [42,43,44,45]. Cyclen was also linked to recognition of the amino acid sequence KLVFF to target and inhibit the aggregation of Aβ peptide [46]. Cyclen increases the solubility of DOPA and provides additional modes of interactions with S100A9. Here, we applied a range of experimental techniques, such as thioflavin-T (ThT)-based amyloid kinetic monitoring, atomic force microscopy (AFM) analysis, and S100A9 titration with the compounds of interest, combining them with computational analysis performed by ligand docking and molecular dynamic (MD) simulation, to draw a broad picture of the inter-molecular interactions between the selected compounds and S100A9, and their effect on S100A9 amyloid self-assembly.

## 2. Results

### 2.1. Computation Analysis of the Physico-Chemical Properties of Molecular Compounds and Their Prospective Drug Likeness

Comparative description of different ligand structures and their general physico-chemical properties may facilitate an analysis of their interactions with S100A9 protein and shed light on their prospective drug likeness [47,48]. Here, we present a summary of the six basic molecular properties of our compounds in the ADME radar diagrams [49], as shown in Figure 1, which depict: ligand flexibility (FLEX), relative sphere of sp^3^ carbon atoms (INSATU), LogP values, or partition coefficient, indicating the concentration of solute in the organic and aqueous partition (INSOLU), polar surface area (POLAR), molecular mass (SIZE), and hydrophobic surface area (LIPO). The pink area in the ADME radar diagram represents the optimal Lipinski values [49], while the superimposed bright red lines represent the values, which are specific for each compound [49]. It is important to note that many of the DOPA and cyclen-based ligands have closely positioned charged groups, which influences the pK_a_ values for each of these groups [47]. The shifts in pK_a_ values were reflected in the calculations of the presented pI values and in the compound net charge at physiological pH 7.2 as illustrated in Figure 1. Most notably, four closely positioned amino groups in the cyclen ring have significant effects on the pK_a_ for each amine in a sequence of protonation events. The calculated pK_a_ values for two secondary amines in cyclen are 9.29 and 8.6, respectively, while the pK_a_ for the third secondary amine and last protonation site in the tertiary amine is 5.41 and 3.0, respectively. Consequently, at physiological pH, the cyclen ring has a net charge of + 2. The three analyzed compounds have pI values close to physiological, while the pI of DOPA-D-H-DOPA and DOPA-D-H shifted to acidic pH due to the negative charges of the DOPA rings.

The prospective drug likeness of the compounds is defined by how much their profiles fall within the optimal region of the ADME radar diagram, though this is a very approximate estimate [49]. Among the five compounds, DOPA-cyclen is characterized by the optimal Lipinski values within the radar diagram, while the four others have some parameters with relatively low deviation from the optimal Lipinski values.

### 2.2. Kinetic Analysis of S100A9 Amyloid Formation in the Absence and Presence of DOPA and Cyclen-Based Compounds Monitored by ThT Fluorescence

The amyloid formation of S100A9 in the absence and presence of DOPA and cyclen-based compounds was monitored by ThT fluorescence and is presented in Figure 2. The amyloid formation correlates with the increase of the ThT fluorescence, and the initial parts of the fibrillation kinetic curves were fitted by using an isodesmic polymerization model [50,51] to derive the kinetic rates; this model has previously been used in kinetic analysis of S100A9 amyloid self-assembly [52,53]. The rates of fibrillation in the presence of the compounds do not deviate significantly from the rate of self-assembly of S100A9 alone (Table 1). An increase of *k* by a factor of two was observed only in the presence of DOPA-H-H-DOPA and DOPA-cyclen at the ratio of S100A9 to compound of 1:10.

Incubation of each individual compound with ThT over the same period of time did not lead to an increase of the ThT signal (Appendix A), indicating that individual compounds do not form amyloids themselves under our experimental conditions.

### 2.3. AFM Imaging of S100A9 Amyloids Aggregated Alone and Together with Corresponding Compounds

In order to complement the kinetic analysis, we also monitored the S100A9 amyloid formation by AFM imaging (Figure 3). The amyloid fibrils of S100A9 alone that developed after 100 h of incubation in PBS, pH 7.4, and 42 °C were very flexible, relatively short, i.e., within 200–600 nm length, and also characterized by a narrow distribution of the heights with a median at ca. 1.6 nm in the AFM cross-sections. In the presence of all DOPA-based compounds, the fibrils of S100A9 become thicker with median heights of ca. 2.0–2.2 nm, and generally longer (Figure 3). In the presence of DOPA-D-H-DOPA and DOPA-D-H, S100A9 fibrils tended to clump together, forming large clustered aggregates, which we have also observed previously upon prolonged aggregation of S100A9 [54]. At the molar ratio of S100A9 to DOPA-based compounds of 1:10, the height of fibrils became even thicker, shifting towards 2.2–2.8 nm median height and in the case of DOPA-D-H-DOPA to 3.0–3.2 nm (Figure 3), while their lengths remained about the same as at the 1:1 molar ratio.

S100A9 fibrils incubated in the presence of H-E-cyclen at the protein to compound molar ratio of 1:1 did not lead to thickening of amyloid fibrils (Figure 3). Some thickening of fibrils with a median of 2.0 nm was observed upon incubation at a 1:10 molar ratio of S100A9 to compound (Figure 3). This indicates that DOPA-based compounds induce more pronounced fibril thickening, even in the presence of cyclen, i.e., in the case of the DOPA-cyclen compound.

### 2.4. Titration of S100A9 with DOPA and Cyclen-Based Compounds Followed by Intrinsic Fluorescence

S100A9 was titrated by each compound of interest following the changes in intrinsic fluorescence intensity, since S100A9 has one aromatic residue: Trp 88. Normalized changes of the fluorescence intensity upon corresponding ligand binding are presented in Figure 4. The fitting was performed using a single binding site model to determine the apparent dissociation constant for each compound. All apparent dissociation constants K_d_ were within the same sub-micromolar range. A twice larger K_d_ was determined only for DOPA-D-H-DOPA binding. The fluorescence spectrum maxima in all complexes of S100A9 dimer with the corresponding compounds were within 342–343 nm, indicating that there were no significant changes in the Trp 88 environment upon ligand binding.

### 2.5. Ligand Docking

The structures of S100A9 homodimer [55] in complex with DOPA and cyclen-based compounds (Figure 1) were described by using a combination of ligand docking and all-atom MD simulation [56]. Molecular docking was used for the initial search of the binding sites of those ligands on S100A9 homodimer [57]. Docking showed that the ligands can bind to three distinct areas on S100A9 homodimer, with the initial limit for potential binding sites set to 9 (Figure 5).

To depict the binding sites and cavities on the molecular surface more clearly, we showed S100A9 molecules in three different presentations, including the ribbon diagrams and two Connolly surface models reflecting the corresponding electric fields [58] and the surface polarities [57]. Only one site, involving the central shallow diagonal groove at the S100A9 homo-dimer interface, accommodates all DOPA and cyclen-based ligands. The diagonal groove is formed by the first 27 amino acid residues of each monomer N-termini, including the first helices (residues 6–27) (Figure 5a). At each end, the groove is limited by the last 20 amino acid residues of each monomer C-termini.

Since the ligands differ in their sizes, charges, and conformations (Figure 1), all these are reflected in their binding to S100A9 dimer. The ligands have a few groups, which can act as H-bond donors and acceptors, while the S100A9 surface is highly polar and charged (Figure 5b,c); therefore, the ligands can form three to six H-bonds upon their binding to the protein surface. In the major site in the central diagonal cavity, only one DOPA-D-H-DOPA binds to this cavity, closely matching its shape (Figure 5). It forms H-bonds with Ser 3, Gln 7, Arg 10, Thr 14, Asn 17, and His 106 in the S100A9 monomer denoted as chain A and with Glu 13 and Thr 14 in the monomer denoted as chain B. DOPA-H-H-DOPA has two binding sites in the S100A9 diagonal cavity, which partially overlap. One site involves Gln 7, Arg 10, and Thr 113 in chain A and Met 5, Gln 7, Arg 10, and Thr 14 in chain B while the second site consists of Glu 13, Thr 14, and Asn 17 in chain A, as well as Gln 7, Arg 10, and Thr 113 in chain B. Therefore, only one DOPA-H-H-DOPA molecule can bind to S100A9 homodimer at a time. DOPA-cyclen and H-E-cyclen are smaller in size (Figure 1), and each of them has two binding sites in the S100A9 diagonal cavity. In the first binding site, DOPA-cyclen forms H-bonds with Gln 7and Thr 14 in chain A and with Arg 10 and Glu 13 in chain B while in the second site, it forms H-bonds with Thr 14 in chain A as well as Lys 93 and Met 94 in chain B. H-E-cyclen forms H-bonds with Arg 10 and Thr 14 of chain A, and Glu 13 and Thr 14 of chain B in one binding site, and with Lys 93 and His 105 of chain A in the second site. By contrast, three DOPA-D-H can bind in the S100A9 diagonal groove, interacting with Thr 14 in chain A and Arg 10 in chain B in one site, Glu 96 and Thr 113 in chain A and Asn 17 in chain B in the second site, and the third site partially overlaps with the second one, including Met 94 and His 95 in chain A.

Other ligand-binding regions on the S100A9 homodimer found by molecular docking are positioned at the interfaces between the second and third helices on each monomer (Figure 5). These sites involve a convex protein surface with a strong positive electric field and no defined cavities. The ligands bound to these sites form only one H-bond in each site. DOPA-H-H-DOPA is the only ligand that does not bind at these sites as it is also positively charged. Dopa-D-H-Dopa and Dopa-D-H tend to bind to only one of these weak binding sites, since only one H-bond is involved in the binding, and slight deviation in the ligand positioning breaks this bond. Thus, few ligands, especially if they are smaller in size, can bind to the S100A9 dimer simultaneously, mostly in the diagonal groove.

### 2.6. MD Simulation

In the MD studies, we analyzed the stability of the binding interactions between the corresponding ligands and S100A9 homodimer by calculating the root mean square deviation (RMSD) values for bound ligands and [59] the number of H-bonds between corresponding ligands and S100A9 (Figure 6).

RMSD values are shown for 100 ns, indicating how long the ligand positioned by molecular docking on the S100A9 surface can remain in the corresponding binding site. The ligands are denoted in order of their mobility on the protein surface as ligands 1 and 2, respectively. If the ligands did not stay on the protein surface longer than 5 ns, their RMSD and H-bonds were not depicted. One molecule of DOPA-D-H-DOPA and DOPA-H-H-DOPA are characterized by RMSD around 5 Å and in general below 10 Å (Figure 6a). They both bind to the diagonal grove and adopt to its conformation well (Figure 5). Another DOPA-H-H-Dopa, which can bind to the diagonal groove, also stays in the binding site over 100 ns, though its RMSD reaches 20 Å. DOPA-D-H-DOPA, which binds to the low-affinity site formed by the second and third helices, does not stay in this site and its RMSD increases to above 25 Å after ca. 40 ns. DOPA-D-H rapidly dissociates from all binding sites as demonstrated by the rapid growth of their RMSD values. Two molecules of H-E-cyclen are characterized by RMSD below 25 Å over 100 ns, indicating that, while they are mobile, they still stay within the diagonal binding cavity. One molecule of DOPA-cyclen binds and stays in the middle of the diagonal cavity over the 100 ns period, displaying RMSD below 25 Å, while the second DOPA-cyclen dissociates from the diagonal groove binding site after ca. 15 ns.

The changes in RMSD values correlate well with the changes in the number of H-bonds between the ligands and S100A9 homodimer (Figure 6b). Docking studies showed that all five compounds form three to six H-bonds with the S100A9 homodimers at the beginning of the MD calculation. MD simulations for one molecule of DOPA-D-H-DOPA and DOPA-cyclen, as well as for both DOPA-H-H-DOPA molecules showed that H-bonds keep the ligand bound to protein over 100 ns, though they can perform small motions within the corresponding binding site (Figure 6). H-bonds keeping other molecules of those types on the S100A9 surface decline to zero after 20 to 40 ns, and molecules become detached from the protein surface. Both molecules of H-E-cyclen maintain H-bonds up till 70–80 ns and then they may dissociate from the S100A9 dimer. By contrast, both molecules DOPA-D-H do not stay bound via H-bonds to the S100A9 surface and may detach after 10–40 ns.

The binding of all DOPA and cyclen-based compounds to the binding sites on S100A9 homodimer derived from MD simulation is depictured in Appendix A. Here, the representative time frames show both details of the ligand binding via H-bonds and van der Waals interactions in the corresponding binding sites (Appendix A) and also zoomed images of the binding sites in 3-D presentation to more precisely locate the binding regions on the homodimer surface (Appendix A). Two ligands (ligand 1 and 2) are depicted in order of their mobility on the S100A9 homodimer surface in accord with the RMSD plot (Figure 6). The MD simulations correlate well with the docking studies (Figure 5). Only the second ligand of DOPA-D-H-DOPA binds to the alternative weak binding site rather than the major groove. All other ligands in order of their mobility bind in the major groove. Four representative videos (Appendix A) demonstrate the mobility of DOPA-D-H-DOPA and DOPA-H-H-DOPA in their corresponding binding sites, respectively, which visually depict what is presented in the RMSD plots (Figure 6). As there are no steric hindrances or significant cavities on the protein surface, ligands are able to move and bounce backward and forward in the binding site, forming multiple contacts over the 100 ns of MD simulation. Interestingly, all DOPA and cyclen-based compounds can interact with the same set of amino acid residues in the major groove, while other binding interactions between polar groups of the compounds and polar protein surface residues may vary significantly and involve multiple H-bond and van der Waals contacts, which form and break due to ligand and protein surface motility.

We compared the RMSD values for one ligand vs. two or more ligands bound to the S100A9 homodimer in the diagonal groove to evaluate whether there is cooperativity in their binding. We found no statistically significant differences in the RMSD or H-bonds regarding whether one or two ligands bind simultaneously in the diagonal groove (data not shown), indicating that there is no cooperativity between consequent ligand binding.

## 3. Discussion

DOPA and cyclen-based compounds have been studied intensively in research targeting amyloid formation of α-synuclein, Aβ, and tau, polypeptides involved in neurodegenerative diseases, and these compounds have shown some potency in amyloid inhibition [28,29,30,31,32,33,34,35]. DOPA is a precursor of the neurotransmitter dopamine and Parkinson’s sufferers have received DOPA as disease-modifying treatment for years [25,26]. Therefore, it was important to examine the effect of DOPA and cyclen-based compounds modified by conjugated amino acid residues on the amyloid self-assembly of the pro-inflammatory protein S100A9, which has been shown to be a central component of the amyloid neuro-inflammatory cascade in Alzheimer’s, Parkinson’s, and traumatic brain injury [11,12,13]. The kinetic analysis of S100A9 amyloid’s self-assembly in the presence of each DOPA and cyclen-based compound taken at both molar ratios of protein to ligand of 1:1 and 1:10 did not show any significant effects on the rate of amyloid formation (Figure 2). This contrasts with previous findings of the inhibiting effects of DOPA and cyclen-based ligands on the amyloid fibrillation of α-synuclein, Aβ, and tau [28,29,30,31,32,33,34,35], indicating that there is no single silver bullet able to inhibit and reverse all the diversity of amyloids. Interestingly, by comparison to Aβ and α-synuclein, S100A9 undergoes amyloid aggregation without a pronounced lag phase, corresponding to a nucleation or oligomerization process, and cannot be seeded by preformed homo or hetero-amyloids [12,15,18,54]. The kinetics of S100A9 amyloid formation are described well by the generic Finke–Watzky autocatalytic model, in which initial protein misfolding and β-sheet formation are defined as the ‘nucleation’ step, spontaneously taking place within individual S100A9 molecules at a higher rate than the subsequent amyloid assembly. Therefore, amyloid self-assembly, described as an autocatalytic process, will proceed if misfolded amyloid-prone S100A9 is populated on a macroscopic time scale [15]. This suggests that dynamic binding of DOPA and cyclen-based derivatives of S100A9 homodimer did not promote S100A9 misfolding, which would consequently affect its amyloid kinetics; however, this effect was not observed here.

Importantly, the AFM-cross-section analysis of co-aggregates of S100A9 with the above compounds, apart from H-E-cyclen, clearly demonstrated that the amyloid fibrils became two to three times thicker compared to S100A9 fibrillated alone (Figure 3). This indicates that the ligand bindings and encapsulation into amyloid fibrils altered the protein conformation and therefore the packing of proteinaceous material at the fibrillar interface, leading to their thickening. Amyloid formation is generally characterized by significant polymorphism, which may be relevant to a specific type of disease and disease development [60,61]. Thicker fibrils may also slow down the tissue propagation of amyloids as it was suggested previously, as the clumping and thickening of amyloid co-aggregates of S100A9 with NCAM constructs has been observed [54]. Thus, analysis of the amyloid fibril morphology and structural studies of amyloid aggregates may provide additional insight into the potential development of the amyloid disease pathology and amyloid propagation [62], which were shown for Aβ peptide [61] and prions [63,64].

All DOPA and cyclen-based ligands were shown to bind to S100A9 homodimer upon titration as monitored by intrinsic fluorescence (Figure 4). Their complex formations were characterized by similar apparent dissociation constants in a sub-micromolar range as determined by fitting the titration curves to a single site binding model. Molecular docking and MD simulation provided additional and important insight into the nature of the binding sites and mechanisms of ligand binding, enabling us to differentiate between the ligand binding modes for each ligand. Indeed, S100A9 is a small protein with no deep cavities on its surface, which makes it a challenging target for drug design [48]. All five studied ligands were docked on the protein surface in the relatively shallow diagonal groove and some additional low-affinity binding sites at the convex interface between the 2^d^ and 3^d^ helices on each monomer (Figure 5). All ligands are characterized by high H-bond donor and acceptor capacities (Figure 1) and are able to make three to six H-bonds with the protein surface at the initial docking (Figure 6). RMSD analysis showed that some complexes with DOPA-H-H-DOPA and DOPA-D-H-DOPA were characterized by less than 5 Å values, indicating their limited mobility and good fit to the binding site. However, other bound ligands were more mobile or even dissociated from S100A9 homodimer if RMSD increased beyond 25 Å. Overall, the binding of various ligands to the major diagonal groove resulted in rather similar apparent K_d_ values as determined in the fluorescence titration experiments.

It is important to note that S100A9 may exist as a homo- and heterodimer, forming complex with S100A8 [14]. The thermal stability of S100A9 homodimer is somewhat lower than S100A8/S100A9 heterodimer as reported previously [65], which may make it more amyloid prone by facilitating its misfolding [15]. Previously, in studies of ex vivo brain tissues in Alzheimer’s and Parkinson’s disease, mild cognitive impairment, and traumatic brain injury, we observed the extra and intracellular amyloid deposits of S100A9, but not S100A8 [11,12,13,66]. Moreover, in mice models of Alzheimer’s disease and traumatic brain injury, we also detected depositions of S100A9, but not S100A8 [16,17]. Furthermore, S100A9 forms amyloid complexes with Aβ, by templating on the Aβ fibrillar surface, which may also contribute to the development of joint amyloid deposits in Alzheimer’s disease [11,18], though these polypeptides do not form a mixed cross-β-sheet [67]. By contrast, in the aging prostate, we observed the co-localization of amyloid deposits of both S100A9 and S100A8 [68]. Even though amyloid development in the brain tissues could be the primary target for DOPA-based compounds, and therefore, here, we studied their interactions with S100A9, it remains an open possibility that in some other functional or pathological conditions in other organ and tissues than the brain, binding of DOPA and cyclen-based compounds to S100A9 may affect its interaction and co-aggregation with S100A8.

It is important to note that DOPA and cyclen-based ligands did not interact with the region on the S100A9 surface, including Lys 50 to Lys 54, which has been shown previously to be critical for S100A9 amyloid self-assembly and by blocking this specific amino acid sequence by polyoxoniobates, we were able effectively hinder S100A9 amyloid growth [53]. However, as the S100A9 amyloid surface is important for its quaternary complex formation and amyloid co-aggregation with Aβ peptide [18] and potentially with other proteins and disease-related and functional amyloids, the amyloid morphology-modifying effect of DOPA and cyclen-based compounds co-aggregated together with S100A9 into fibrils should be taken into account in future studies of protein hetero-aggregation.

## 4. Materials and Methods

### 4.1. Amyloid Fibril Formation

S100A9 protein was expressed in *E. coli* and purified as described previously [65]. Freshly dissolved S100A9 in PBS, pH 7.4 was used in all measurements. Lyophilized S100A9 was dissolved on ice in PBS buffer at pH 7.4. Then, S100A9 samples were filtered using a 0.22 µm spin membrane filter to remove any preformed aggregates. These solutions were directly subjected to experiments. To produce amyloids, S100A9 was incubated in PBS, pH 7.4, and 42 °C. The cyclic compounds (Figure 1) were dissolved in PBS at pH 7.4, and their stock solutions were kept frozen prior to measurements.

### 4.2. ThT Fluorescence Assay

ThT dye is known to bind specifically to β-sheet-containing amyloid structures and thus enables quantification of the kinetics of amyloid self-assembly. ThT assay was performed as described previously [69]. First, 75 μM S100A9 was transferred into Nunc 96U black well plates with transparent bottoms and then 20 µM ThT was added to each well. The compounds dissolved in PBS were added to S100A9 solution with molar ratios of 1:1 and 1:10, respectively. Sample volumes were kept at 200 μL per well. The plates were immediately covered, placed into a Tecan F200 PRO plate reader, and incubated at 42 °C with 432 rpm orbital shaking every 10 min. ThT fluorescence was recorded every 10 min. Filters at 430 and 495 nm wavelengths with a 20 nm band width each were used for excitation and emission, respectively. Each sample was incubated in triplicate. ThT fluorescence intensities were normalized.

### 4.3. Kinetic Curve Fitting

An isodesmic polymerization model was used to fit the kinetic dependencies of S100A9 amyloid formation monitored by ThT fluorescence as described previously [53], since these curves displayed hyperbolic dependences characteristic of this model [45,46].

### 4.4. AFM Imaging

AFM imaging was performed by using a BioScope Catalyst AFM (Bruker; Phoenix, AZ, USA), operating in a peak force mode in air. The scan rate was 0.51 Hz and the resolution was 512 × 512 pixels. Bruker MSLN and SLN cantilevers were used in all measurements. Imaging was also conducted using a PicoPlus AFM (Molecular Imaging), equipped with a 100 μm scanner, operating in tapping mode in air. The resonance frequency was set in 170 to 190 kHz range, scan rate of 1 Hz, and resolution of 512 × 512 pixels. For ambient imaging, 20 µL of each sample were diluted 200× in deionized water, deposited on the surface of freshly cleaved mica, kept for 15 min, washed 4 times with 200 µL of Milli-Q deionized water, and left to dry overnight at room temperature. The height distributions of amyloid fibrils were measured by a Bruker Nanoscope Analysis v. 1.5 cross-section tool for at least 50 fibrils for each sample.

### 4.5. Titration of S100A9 with DOPA and Cyclen-Based Compounds Monitored by Intrinsic Fluorescence

First, 1 μM S100A9 was incubated in PBS, pH 7.4, room temperature for 3 h in the absence or presence of 5 nM to 2.4 μM of the studied compounds prior to fluorescence measurements. The final volume was kept at 400 μL for all samples. The fluorescence emission spectra were recorded using a Jasco spectrofluorometer FP 6500 (Japan), setting the excitation at 296 nm and emission at 350 nm wavelengths, respectively, with a 3 nm band-width on both the excitation and emission. The measurements were performed in triplicates. Fitting of the normalized titration curves was performed by using a single site binding model.

### 4.6. Ligand Docking Studies

The S100A9 structure used in this study was the average from a set of 10 NMR structures downloaded from the protein data bank, ref PDB: 5I8N [55]. The ligands were hydrogenated and charged at pH 7.2, using the Gasteiger protocol [56]. S100A9 was protonated at pH 7.2 using the AMBER98S force field [57]. Ligand docking studies started with a search box that enclosed the entire protein [57].

### 4.7. All-Atom MD Studies

S100A9 homodimers in complex with corresponding ligands prepared in molecular docking studies were used as starting structures for MD simulations [70], including from one to nine bound ligands. These complexes were prepared for MD calculations using CHARMM-GUI solution builder [71]. Ligand parametrization used the CHARM36 force field with LigPrep protocols [71]. In a typical calculation, protein-ligand complex was placed in a water box that had about 132,000 atoms. TIP3 models for water molecules were combined with sodium and chloride ions adjusted to a concentration of 150 mM and the net charge set to zero. The system relaxation used a sequence of equilibration steps at 303.15 K with Nose–Hoover coupling, and the pressure was set to 1.0 bar with semi-isotropic Parinello–Rahman coupling. The Verlet cutoff scheme was combined with the LINCS constraint algorithm. System relaxation used two minimization steps, which were followed by one equilibration step. The simulations analyzed the molecular processes for 100 ns, containing 50 million steps with the step size set to 2 fs. All simulations were run on GROMACS 2020.4 version [59] for 32 h on 240 cores with 480 logical cores.

### 4.8. DOPA and Cyclen Ligand Synthesis

DOPA and cyclen-based compounds were synthesized as described previously [72].

## Figures and Tables

**Figure 1 ijms-22-08556-f001:**
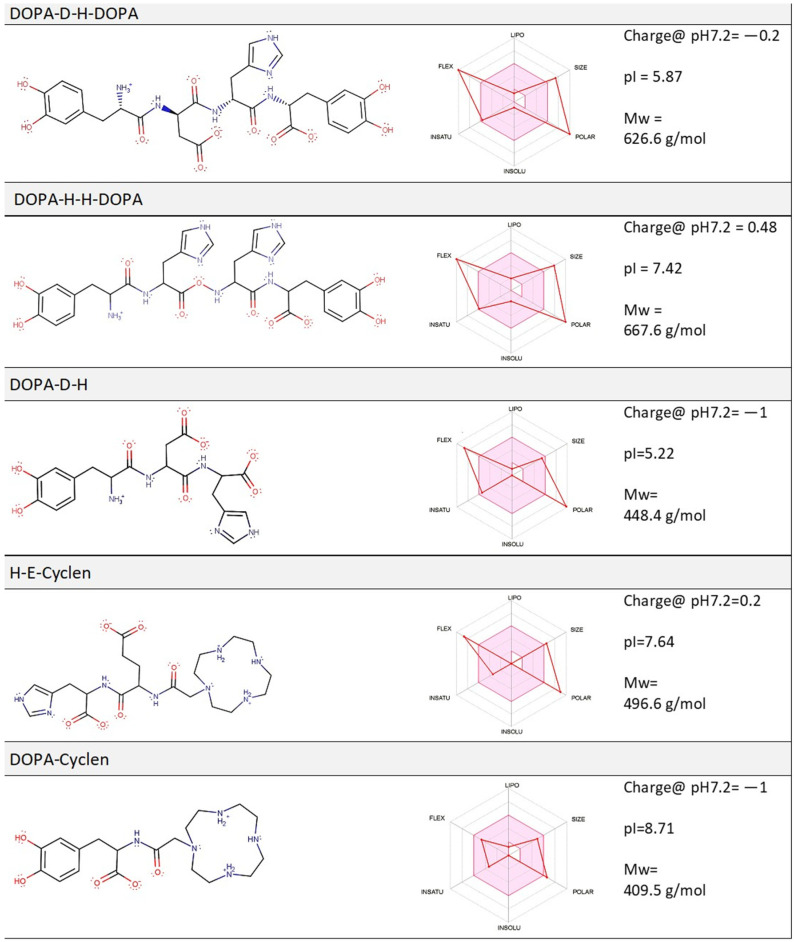
Schematic presentation of chemical structures and physico-chemical properties of selected compounds: DOPA-D-H-DOPA, DOPA-H-H-DOPA, DOPA-D-H, H-E-cyclen, and DOPA-cyclen. Schematic presentations of molecules are shown combined with ADME radar diagrams, depicting their six physico-chemical properties and drug likeness, i.e., how much the molecular properties fall into the pink area.

**Figure 2 ijms-22-08556-f002:**
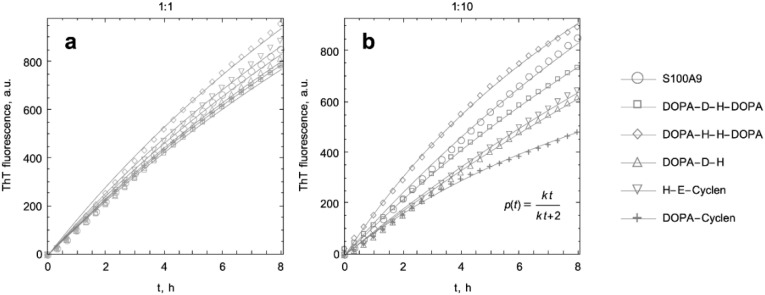
Kinetics of S100A9 amyloid formation in the absence and presence of DOPA and cyclen-based compounds monitored by ThT fluorescence. (**a**) Amyloid kinetics monitored at the ratio of S100A9 to corresponding compound of 1:1 and (**b**) 1:10, respectively. Symbols indicating the experimental data points for S100A9 alone or with respective compounds are shown in the caption and the corresponding lines indicate the fitting with the isodesmic polymerization model. The equation for isodesmic polymerization fitting is shown in ((**b**) inset). 75 μM S100A9, PBS, pH 7.4, and 42 °C.

**Figure 3 ijms-22-08556-f003:**
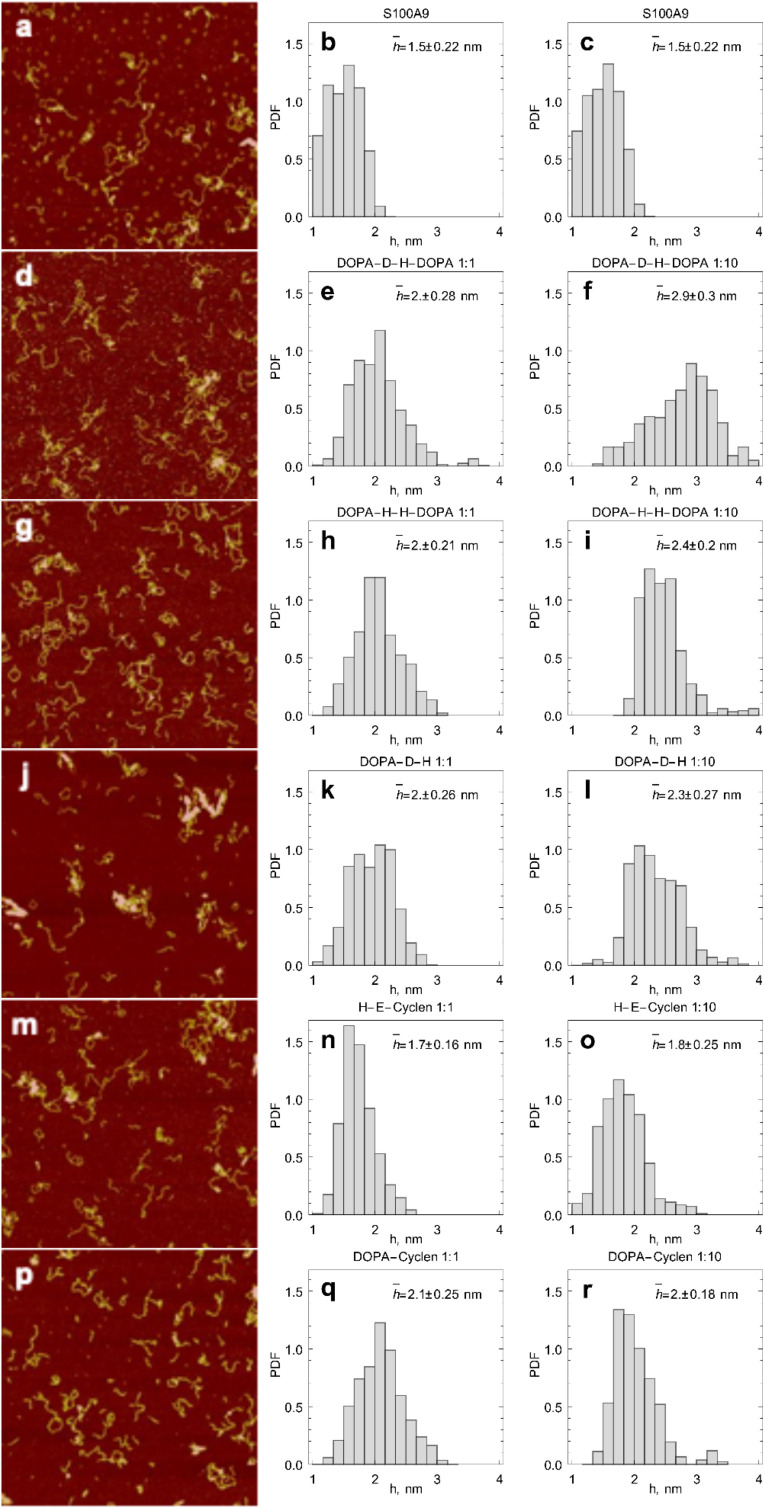
AFM imaging of S100A9 fibrils in the absence and presence of DOPA and cyclen-based compounds and the analysis of fibril height distributions. (**a**) Representative AFM image of S100A9 fibrils incubated alone. (**b**,**c**) Distribution of fibril heights measured in AFM cross-sections. The same distribution is shown in both columns to facilitate the comparison with the following height distributions carried out in the presence of compounds. (**d**,**g**,**j**,**m**,**p**) Representative AFM images of S100A9 fibrils in the presence of DOPA and cyclen-based compounds at a 1:1 molar ratio of S100A9 to compound. (**e**,**h**,**k**,**n**,**q**) Distributions of fibril heights measured in AFM cross-sections at the S100A9 to compound molar ratio of 1:1 and (**f**,**i**,**l**,**o**,**r**) at molar ratio of 1:10. Image sizes are 2.5 × 2.5 μm. The compounds and molar ratios are indicated above the figures. Samples were incubated for 50 h in PBS, pH 7.4, and 42 °C.

**Figure 4 ijms-22-08556-f004:**
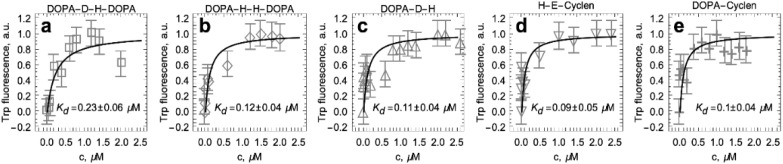
Binding of DOPA and cyclen-based compounds to native S100A9 followed by intrinsic fluorescence. (**a**–**e**) Normalized titrations of S100A9 by corresponding compounds. Each compound is indicated above corresponding figure (**a**–**e**) and their K_d_ in captions. Symbols indicate the experimental data points and lines the fitting with a single binding site model. 75 μM S100A9, PBS, pH 7.4, and 20 °C.

**Figure 5 ijms-22-08556-f005:**
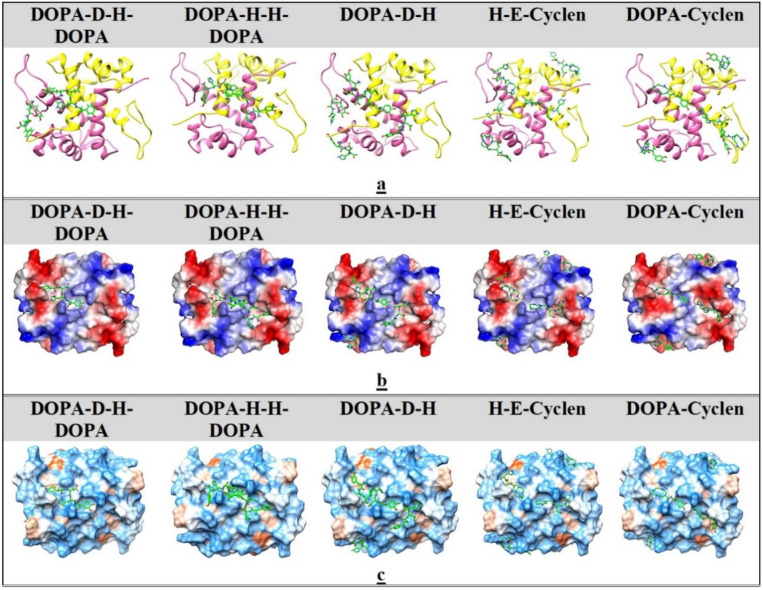
Binding sites on S100A9 homodimer for corresponding DOPA and cyclen-based compounds derived from molecular docking studies. (**a**) Ribbon diagrams of S100A9 homodimer with corresponding bound compounds, depicting the positions of the bound ligands on their surface. S100A9 monomers are depicted as yellow and pink ribbons and denoted as chains A and B, respectively. The ligands are depicted by green sticks in all images in (**a**–**c**). (**b**) Connolly surface electric potential models of S100A9 homodimers with corresponding bound compounds. Electric potentials are depicted on scale from −8 to 8 k_B_T/e in red-white-blue colors (red shows the negative extreme in electrostatic potential, blue the positive extreme, and white the non-polar regions). (**c**) Connolly surface polarity models of S100A9 homodimer with the corresponding bound compounds depicting hydrophilic (in light blue), intermediate (in white), and hydrophobic (in orange) patches on its surface. The bound compounds are indicated in the figure caption above the corresponding images.

**Figure 6 ijms-22-08556-f006:**
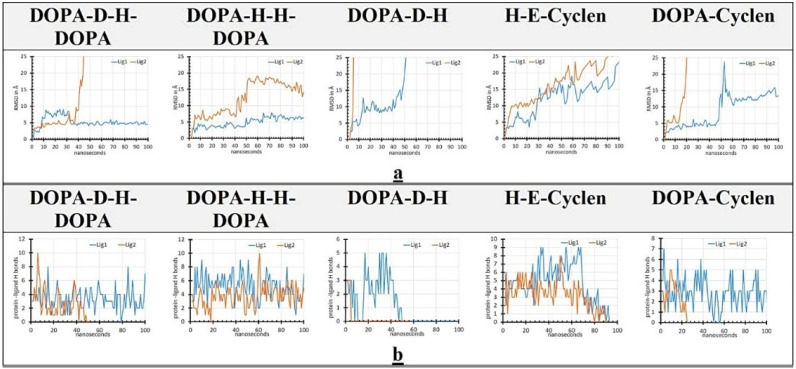
All-atom MD simulations of the binding interactions between S100A9 homodimer and DOPA and cyclen-based compounds. (**a**) RMSD values depicting the corresponding compound mobility in complex with S100A9 dimer over 100 ns. The ligands denoted as 1 and 2 are presented in order of their mobility on the protein surface; the least mobile ligand 1 is shown in blue and the more mobile ligand 2 is shown in orange, respectively. (**b**) Number of H-bonds between the corresponding ligands and S100A9 dimer. The color coding in (**b**) is the same as in (**a**).

**Table 1 ijms-22-08556-t001:** Kinetic rate constants of S100A9 amyloid formation in the present of compounds derived from the kinetic experiments monitored by ThT fluorescence.

Protein and Compounds	Kinetic Rate Constants and Their Rations			
	1:1		1:10	
	k, μM^−1^s^−1^	k/k_S100A9_	k, μM^−1^s^−1^	k/k_S100A9_
S100A9	0.066	1	0.066	1
DOPA-D-H-DOPA	0.06	0.91	0.066	1
DOPA-H-H-DOPA	0.077	1.17	0.134	2.03
DOPA-D-H	0.063	0.95	0.045	0.69
H-E-cyclen	0.069	1.05	0.048	0.72
DOPA-cyclen	0.062	0.93	0.133	2.02

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
