# Peer review of "Co-Aggregation of S100A9 with DOPA and Cyclen-Based Compounds Manifested in Amyloid Fibril Thickening without Altering Rates of Self-Assembly"

_ijms, 2021, doi:10.3390/ijms22168556_

Round 1

Reviewer 1 Report

This is an interesting and important study that adds significantly to the field and will have a noticeable impact. The manuscript is well-written and concise. There are two issues that need to be addressed.

1) It is known now that amyloids are not only pathological troublemakers, but might have important functional roles. Although this observation is not related to the major theme of this manuscript tat looks at the pathological fibrillation, in my view, the authors should add a short paragraph on the abundance and importance of functional amyloids.

2) The authors implied that DOPA and Cyclen-based compounds can interact with S100A9. Support for the actual existence is given in the form of intrinsic fluorescence measurements and computational docking and MD studies. It would be great to see some other evidence of S100A9 interaction with DOPA and Cyclen-based compounds. Can the authors measure near-UV CD spectra of S100A9 in the absence and presence of these compounds?      

Reviewer 2 Report

In this paper, Arabuli et al describe the binding properties and effects on amyloid formation of various DOPA- and cyclen-derived compounds with respect to the neuroinflammatory S100A9 protein. These compounds were previously shown to inhibit the formation and/or reduce the toxicity of Aβ, tau and α-synuclein aggregates. To evaluate the full therapeutical potential of these compounds, the authors here investigate their effects on another prone-to-aggregate protein found in the brain, S100A9, which is a key inducer of neuroinflammation during Alzheimer’s or Parkinson’s diseases. Using intrinsic fluorescence measurements, ligand docking and MD simulations, they show that all tested compounds can bind to S100A9, primarily via a groove located at the subunit interface within the S100A9 homodimer. They also demonstrate that the compounds do not inhibit the formation of S100A9 fibrils neither affect the kinetics of aggregation. Instead they lead to thicker amyloids. Whether this might slow down the neurodegenerative process and therefore be an advantage for therapeutical purpose remains to be determined.

The findings of the authors are quite interesting since they show that DOPA/cyclen compounds have a distinct effect on S100A9 as compared to other aggregative proteins. This raises novel questions on their potential use as therapeutical agents in neurodegenerative disorders. The data presented are sound, the methodologies are well documented and the conclusions are appropriate. I would therefore recommend this article for publication in IJMS, providing that a few concerns listed below are addressed by the authors.

1) Visualization of the groove in which all ligands are docked in Figure 5 is not easy. A zoomed panel with annotations on the structure to label the helices and key residues should be added to locate more precisely this region on the homodimer surface. Such figure would also help follow the description of the putative H-bonds formed between DOPA-compounds and S100A9 (lines 240-260).

2) Similarly, a zoomed panel should be included to visualize more precisely the second binding groove described page 9 (lines 260-270).

3) The description of H-bonding between S100A9 and the compounds is quite detailed although it is only based on docking studies. Mutational studies coupled to intrinsic fluorescence measurements should be performed on a few key residues to validate the binding sites identified by docking.

4) In the introduction, the authors mention that some DOPA-derived compounds have been shown to form amyloid-like assemblies. On the other hand, when testing the Tht fluorescence of the DOPA- and cyclen-based compounds alone, they do not see any changes in fluorescence but these data are not shown in the paper. I am not a big fan of the “data not shown” concept. Either the data are of no interest for the paper. Then, no need to mention them. Or they are of interest, or they provide a valuable control/reference. In that case, they should be displayed, at least in supplementary. In conclusion, I think the ThT fluorescence signals recorded for the compounds alone should be displayed in the manuscript, at least in supplementary.

5) Does binding of the compounds to S100A9 affect the oligomeric state of the protein in conditions where amyloid would not form (e.g. a few hours incubation at room temperature)? Could the compounds be involved in bridging together two S100A9 homodimers? Size exclusion chromatography experiments would help answer this question. Similarly, how do divalent cations, in particular calcium, influence the binding of the DOPA/cyclen-compounds to S100A9?

6) Page 11, lines 360-363 : « Overall, the variable number of bound ligands (larger number of smaller size ligands) and their different affinity in the diagonal groove and in the additional low affinity binding sites resulted in rather similar apparent Kd determined in the fluorescence titration experiments ». It is not clear what the authors mean by this sentence.

My conclusion would rather be that in the intrinsic fluorescence titration experiments, the apparent Kd measured corresponds to binding of 1 molecule to the highest affinity binding site (most probably the diagonal groove) and the other low affinity sites are not detected, which is consistent with best fit with a one binding site model. This paragraph and the overall conclusions or at least hypotheses of the authors on how many ligands can bind simulatneously to S100A9 should be clarified.

7) Many studies have suggested that in vivo, S100A9 would be mostly present as form of a heterocomplex with S100A8, the respective homo-complexes being much less stable.

  1. Is this also the case in the brain?
  2. Are S100-based fibrils composed only of S100A9 in vivo? Or do they also contain S100A8?
  3. Would the DOPA/cyclen-based compounds described in this study also bind to S100A8? Or to the S100A8-A9 heterodimer?

Minor comment:

- the manuscript should be carefully proofread, there are quite a few spelling mistakes throughout the text
